# Site Mutation Improves the Expression and Antimicrobial Properties of Fungal Defense

**DOI:** 10.3390/antibiotics12081283

**Published:** 2023-08-03

**Authors:** Ya Hao, Da Teng, Ruoyu Mao, Na Yang, Jianhua Wang

**Affiliations:** 1Gene Engineering Laboratory, Feed Research Institute, Chinese Academy of Agricultural Sciences, 12 Zhongguancun Nandajie St., Haidian District, Beijing 100081, China; 2Innovative Team of Antimicrobial Peptides and Alternatives to Antibiotics, Feed Research Institute, Chinese Academy of Agricultural Sciences, 12 Zhongguancun Nandajie St., Haidian District, Beijing 100081, China; 3Key Laboratory of Feed Biotechnology, Ministry of Agriculture and Rural Affairs, Beijing 100081, China

**Keywords:** antimicrobial peptides, recombination expression, degenerate bases, DNA library, *Staphylococcus aureus*

## Abstract

Although antimicrobial peptides (AMPs) have highly desirable intrinsic characteristics in their commercial product development as new antimicrobials, the limitations of AMPs from experimental to scale development include the low oral bioavailability, and high production costs due to inadequate in vitro/in vivo gene expression- and low scale. Plectasin has good bactericidal activity against *Staphylococcus* and *Streptococcus*, and the selective bactericidal activity greatly reduces the damage to the micro-ecosystem when applied in vivo. However, its expression level was relatively low (748.63 mg/L). In view of these situations, this study will optimize and modify the structure of Plectasin, hoping to obtain candidates with high expression, no/low toxicity, and maintain desirable antibacterial activity. Through sequence alignment, Plectasin was used as a template to introduce the degenerate bases, and the screening library was constructed. After three different levels of screening, the candidate sequence PN7 was obtained, and its total protein yield in the supernatant was 5.53 g/L, with the highest value so far for the variants or constructs from the same ancestor source. PN7 had strong activity against several species of Gram-positive bacteria (MIC value range 1~16 μg/mL). It was relatively stable in various conditions in vitro; in addition, the peptide showed no toxicity to mice for 1 week after intraperitoneal injection. Meanwhile, PN7 kills *Staphylococcus aureus* ATCC 43300 with a mode of a quicker (>99% *S. aureus* was killed within 2 h, whereas vancomycin at 2× MIC was 8 h.) and longer PAE period. The findings indicate that PN7 may be a novel promising antimicrobial agent, and this study also provides a model or an example for the design, modification, or reconstruction of novel AMPs and their derivatives.

## 1. Introduction

To address the antibiotic resistance crisis, new antimicrobial agents and mechanisms of action are essential for finding more new drug candidates [1]. Antimicrobial peptides (AMPs), as a promising new class of antibacterial molecules, are produced by organisms from all areas of life and constitute an almost first universal defense mechanism against infectious agents with a wide range of activities to kill bacteria, fungi, protozoon, tumor and improve animal immunity [2,3]. The bactericidal mechanisms of AMPs are different from traditional antibiotics. The latter acts by eliminating functions necessary for microbial growth or survival, such as blocking the synthesis of bacterial proteins or altering enzyme activity to kill bacteria, while bacteria can cope with the attack by changing a gene. Most AMPs act on bacterial cell membranes, resulting in increased membrane permeability so as to penetrate and kill bacteria. The bacteria would have to change the structure of the membrane, that is, change a significant portion of the genes, to defend against the AMP attack, and this is almost impossible in bacteria. Therefore, AMPs greatly reduce the possibility of resistance developing [4,5,6]. Rapid antimicrobial action is an obvious advantage of AMPs over traditional antibiotics, and it is also one of the reasons why AMPs are not easy to induce bacterial resistance [7,8]. Another advantage of AMPs over antibiotics is that they tend to be less stable and much less environmentally persistent than antibiotics, despite the former limits or shortens necessary action period in vivo showing like a double-edged sword. The persistence of drugs in the environment at sublethal concentrations constitutes an important driver of antibiotic resistance evolution [9]. These reasons make bacteria less susceptible to drug resistance when AMPs are used [10]. Therefore, AMPs, as one of the ideal antimicrobial infection drugs, have become a research and development hotspot in recent years since the new century.

To date, many AMPs are in or have been stopped in the stages of clinical trials of new drugs, and relatively few AMPs have successfully entered the market and application, showing the difficulty that needs to be overcome and the long way to go [3,11,12]. The DRAMP (data repository of antimicrobial peptides) summarizes 96 AMPs in drug development; for example, Gramicidin S/D has completed clinical trials and was released into the market [13]. AMPs have had several challenges moving from initial evaluation to clinical usage [14,15]. Natural AMPs have a series of problems, such as complex biological activities, high toxicity, and low stability and yield, which seriously limit and hinder their industrial application [16,17]. With advances in optimization strategies and molecular design, there is great potential to improve the physicochemical and biological properties of AMPs. Current methods for optimizing existing AMPs and designing new AMPs to increase potency or selectivity include rational design, high-throughput screening, and computational-aided design [12,18]. High-throughput screening can significantly improve the discovery efficiency of AMPs and expand the amount of data available on antimicrobial activity, revealing potential associations between peptide sequences/structures and biological activity against target pathogenic bacteria [19]. The use of gene-coding peptide libraries in screening results in unique and synthetic AMPs. Ritter et al. designed a library of 960 microcin J25 mutants to screen for ones with specific antibacterial activity against pathogenic *Salmonella* [20]. The PCR-based techniques, such as DNA shuffling [21] and site-saturation mutagenesis [22,23], in which randomly generated nucleic acid libraries encoding for AMPs are expressed in a host, result in a library with a huge abundance and complexity, which can easily operate peptide molecules in different expression systems [24].

Animal-derived antimicrobial peptides are synthesized in small amounts in response to stress and are useful for fine-tuning immunity [25,26], but their druggability is relative weak, and thus their development direction should be focused on immune regulation as expected. The microbial-derived peptides that they are more close to peptide antibiotics especially for the narrow-spectrum antibiotics as vancomycin, nosiheptide, polymyxin-, daptomycin and others, have good clear antibacterial mechanism and drug-forming properties, i.e., druggability, which is more suitable for development as a new source of candidate antibacterial drugs, and were produced by synthetic biology including gene and metabolic engineering and gene cluster transfer by the present feasible way [26,27,28,29,30]. Plectasin is the first fungal defensin that was isolated from the secreted protein of *Pseudoplectania nigrella* by Mygind and co-workers in 2005 [31]. Its structure and mechanism of action are clear, and no cross-tolerance exists between Plectasin and traditional antibiotics [32,33]. Plectasin acts on cell wall biosynthesis by targeting the precursor of bacterial cell wall Lipid II. Plectasin does not inhibit the activity of the enzyme involved in cell wall synthesis but binds to Lipid II at a 1:1 stoichiometric ratio. NMR analysis showed that F2, G3, C4, and C37 residues interact with the pyrophosphate moiety of Lipid II via hydrogen bonding [33]. Plectasin has particularly strong bactericidal activity only against Gram-positive bacteria, especially *Staphylococcus* and *Streptococcus* [31]. The selective bactericidal activity greatly reduces the undifferentiated attack to the microecosystem of the treaters when it is applied in vivo [34,35,36]. Evaluating its merits from the above selectivity is difficult depending on different personnel expectations. However, its expression level was relatively low (748.63 mg/L) [37], and an improvement increment of 5–10 times would be reasonably expected for its purpose at an industrial scale or level in the future. In view of these actual situations, development expectations, and our previous experience in similar directions over the years, the structure of Plectasin will be modified and optimized in this study to obtain candidates with high expression and antibacterial activity, no/low toxicity, and better druggability. A challenge has been how to use AMP databases in the actual field to design and construct new AMPs [3,38,39]; thus, our aim would try to follow and solidate a clue from the AMP databases for new candidate derivatives by enhancing the combination of theory and application.

## 2. Results

### 2.1. Sequence Design and Screening

The sequence alignment showed that amino acids differed at several sites, i.e., 9, 11, 13, 32, and 33 (Figure 1). We designed three sequences containing different numbers of degenerate bases (Table 1) according to the final comprehensive desirable aims by merging all or different characteristics in the target peptide involving the physical, chemical, bio-activity, biosynthesis, biosafety and stability characteristics together. Using DNA recombination and high throughput screening, better Plectasin variants could be found. The study performed three rounds of inducible expression and screening at different levels: well plates, shake flasks, and 5 L fermenters. The detection methods of each round were inhibitory zone, minimum inhibitory concentration (MIC), and protein yield, respectively (Figure 2). In total, 3000 active transformants were obtained using an inhibitory zone experiment. Figure 3 illustrates part of the inhibition zones for well plate screening. The transformants with good inhibitory zone effects were selected for sequencing and nine sequences (PN1–PN9) were determined (Table 2). Sanger sequencing showed that the amino acids of the nine peptides screened were all different at the 9th, 13th, and 14th site. Based on the initial analysis of the above results, these three sites may be recognized as the key sites affecting the nature of Plectasin for drug development.

The MIC was measured after purification of the PN1–PN9 protein induced by shaking the flasks. Except for PN9, all other sequences showed good antibacterial activity with MIC values ranging from 4 to 32 μg/mL (Table 3). The recombinant expression vectors of PN1–PN9 were constructed, the optimal transformants were screened, and finally, three transformants PN5, PN6, and PN7 were selected for high-density expression in a 5 L fermenter. The inhibition zones detected the activity of the protein in the supernatant of fermentation. The results showed that PN5 and PN7 had the highest activity at 120 h of induction (Figure 4A–C). As can be seen from Figure 4D–F, the total protein yield of PN5, PN6, and PN7 reached the highest, about 1.85 g/L, 2.59 g/L, and 5.53 g/L (supernatant), and the cell wet weight was about 331.78 g/L, 370.30 g/L, 390.96 g/L after 120 h of induction, respectively (Figure 4D–F).

PN7 showed the highest recombinant yield and was used as the best candidate for further study of the physical and chemical properties and antibacterial effects in vitro. The relative molecular weight of the purified protein identified by MALDI-TOF MS was 4409.78 Da, which was equivalent to the theoretical MW value of 4404.54 Da (Figure 4G). The additional ~5 kDa was considered as a result of the non-specific cleavage of the protease Kex2 which cut the signal peptide, and another two peaks of 8808.697 Da and 13211.203 Da were probably from the polymerization of the target peptides under high concentration situations (Figure 4G).

### 2.2. Structure Analysis of PN7

To analyze the structural features of PN7 and Plectasin, the circular dichroism (CD) spectra of the peptide were measured in ddH_2_O, sodium dodecyl sulfate (SDS), and 2,2,2-Trifluoroethanol (TFE) buffer, respectively. The secondary structures of PN7 in ddH_2_O were characterized predominantly by antiparallel with a positive peak near 195 nm and a negative peak between 210 and 220 nm (Figure 5A), indicating that the secondary structure of PN7 is dominated by β-sheet. The CD spectra showed an increase in the α-helical content of PN7 in SDS and TFE compared with the one in ddH_2_O (Table 4). This result indicated that both the SDS and TFE are favorable to forming α-helical structures to support better antimicrobial functions. The CD spectrum trends are similar for PN7 and Plectasin (Figure 5, Table 4 and Table 5).

### 2.3. PN7 Had Potent Antimicrobial Activity

As shown in Table 6, PN7 showed the high activities against both *Streptococcus* and *Staphylococcus*. The MIC values were in a range of 1 to 16 μg/mL. For *S. aureus* CVCC 546, *S. aureus* E48, *Streptococcus. suis* CVCC 606 and *S. agalactiae* CAU-FRI-4, the MIC values are 1 μg/mL. The PN7 had the strong antimicrobial activities to *S. aureus* CICC 10473, *S. hyicus* NCTC 10350, *S. aureus* ATCC 43300, and *S. hyicus* 437-2 with MICs of 4–8 μg/mL. Meanwhile, it showed *S. aureus* ATCC 25923 and S. epidermidis ATCC 35984 with same MICs of 16 μg/mL. However, PN7 showed very weak activity against Gram-negative bacteria such as *Escherichia coli*, *Salmonella*, and *Pseudomonas aeruginosa* with MIC values over 128 μg/mL. Similarly, the MIC value of PN7 against *Candida albicans* CICC 98001 is also greater than 128 μg/mL. This narrow spectrum would provide direct support for the treatment of Gram-positive bacteria in future clinical practice.

### 2.4. In Vitro and In Vivo Toxicity of PN7

As can be seen from Figure 6A, PN7 showed very low hemolysis against mouse erythrocytes. Even at a high concentration of 128 μg/mL, the hemolysis of PN7 was only 0.19%. The cell viability of PN7 against mouse macrophages RAW264.7 and human keratinocyte HaCaT were 92.77% and 85.36% at 128 mg/mL, respectively, significantly higher than that of Plectasin (67.18% and 62.85%) (Figure 6B,C). This suggested that PN7 had a low cytotoxicity against eukaryotic cells. Meanwhile, we assessed the in vivo toxicity of PN7 in CD-1 mice through daily intraperitoneal injection of PN7 at a dose of 10 mg/kg for 6 days. During one week of intraperitoneal injection, we did not observe a change in murine behavior or the loss of body weight (Figure 6D). Serum biochemical index and whole-blood cell profile detection showed similar levels to the control group (Appendix A). At the same time, the pathological sections of the mouse organs’ histological morphology showed no difference between them (Figure 6E). These solidify the safety basis of PN7 as drug candidate.

### 2.5. Desirable Stability of PN7

As shown in Figure 7, PN7 was stable at 20–80 °C and pH from 2 to 10, while at 100 °C, its antibacterial activity decreased by 1-fold (Figure 7A,B). Furthermore, the addition of monovalent cations (Na^+^ and K^+^) had no influence on the activity of PN7, while the existence of high divalent cation Mg^2+^ and Ca^2+^ strength impaired highly the activity of PN7 with MIC values increased by 1 to 7-fold, but the trace amount showed no effect (Figure 7C,D). Interestingly, PN7 conserved full activity after incubation with simulated gastric juice (SGF), while simulated intestinal fluid (SIF) abolished the activity of the PN7 peptide (Figure 7E); PN7 had a strong resistance to the serum effect (Figure 7F). Overall, these data suggested that PN7 showed good thermal, pH, and serum stability. These paved and validated the basic valuable information for decision-making for selection on the drug form and administration mode.

### 2.6. In Vitro Antimicrobial Analysis of PN7

#### 2.6.1. Time-Killing Curves

*S. aureus* ATCC 43300 was cultured to the logarithmic stage, MHB medium was used for adjusting the concentration of bacterial suspension, and PN7 (1×, 2× and 4× MIC), Plectasin (1×, 2× and 4× MIC) and 2× MIC vancomycin (the MIC was 1 μg/mL) were added into the suspensions containing 10^5^ CFU/mL bacteria (after the count of viable bacteria detection, the initial inoculation amount was 5.15 Log10 CFU/mL), cultured at 37 °C and 250 rpm. A 100 CFU/mL is considered the detection limit. As shown in Figure 8A, after treatment with 1×, 2× and 4× MIC PN7, the counts of *Staphylococcus aureus* ATCC 43300 were reduced by 2.25, 2.42, and 2.72 Log10 CFU/mL during the first two hours of treatment, respectively. In contrast, the count was reduced by 1.6 Log10 CFU/mL after treatment with 2× MIC vancomycin, an approximately 66% population decrease of target pathogens compared with the PN7 treatment group at the same time point of the second hour. After 4 h, the colony number of the peptide treatment group was almost 0, while that of the vancomycin treatment group was 0 after 8 h, at 4 h later than the PN7 treatment. At 24 h, the number of colonies of only 1× MIC Plectasin treatment group rebounded, while no such phenomenon occurred in the other treatment groups. The results showed that PN7 could quickly kill *S. aureus* ATCC 43300 at the initial period of treatment, and its quick bactericidal activity was superior to Plectasin and vancomycin; it will meet requirements for quick-killing of pathogens during clinical practice.

#### 2.6.2. The Post Antibiotic Effect (PAE) of PN7 against *S. aureus*

Both PN7 and vancomycin showed increased PAE in a dose-dependent manner. The PAE values of PN7 to *S. aureus* ATCC 43300 were 1.32, 1.97, and 2.74 h at 1×, 2× and 4× MIC, respectively, which was significantly longer than that of vancomycin (0.11, 0.54, and 0.78 h) (Figure 8B). A big gap of 12, 3.648, and 3.513 times of PAE from the above three dosages showed the advantage of the long effective period from PN7 and would support a huge decrease in effective dosage or administration times during clinical treatment.

#### 2.6.3. Morphological Observation

As shown in Figure 8C, the untreated *S. aureus* cells exhibited an intact cell morphology and smooth surface via scanning electric microscopy (SEM) observation. After treatment with 2× MIC PN7 for 30 min, shrinkage only appears in the cell surface of *S. aureus* ATCC 43300. Finally, a serious disruption occurred and was found in *S. aureus* cells with time. The same result was observed after 2× MIC Plectasin treatment. Of course, more details should be designed, treated and revealed by means of stronger cell morphological technic support.

#### 2.6.4. Effect of PN7 on Membrane Penetrating

In the absence of peptides, the percentage of *S. aureus* ATCC 43300 cells stained with PI was only 0.54%, indicating well-integrated cell membranes. The PI-staining percentage of *S. aureus* ATCC 43300 cells treated with 2× MIC PN7 for 30 min, 60 min, and 90 min was 16.7%, 30.4%, and 34.4%, respectively (Figure 8D). A dose-dependent increase in PI fluorescence indicated that PN7 can damage the cell membranes of *S. aureus* ATCC 43300 with peptide concentration. After treatment with 2× MIC Plectasin, the PI-positive percentage was 12.0% (30 min), 25.2% (60 min), and 32.0% (90 min). The former PN7 showed a slightly stronger membrane penetrating ability than the latter; however, their tendencies are consistent with the morphological observation from the SEM.

## 3. Discussion

Although antimicrobial peptides have highly desirable intrinsic characteristics in new drug development, and the number of patents on novel AMPs has increased significantly over the past 10 years, relatively few have reached the final stages of clinical trials. Limitations or discontinuous ending of AMPs from experimental to application development are due to many factors, including a low oral bioavailability of AMPs formulations, poor stability, and druggability, as well as a lack of high expression or synthesis system for AMPs, and high production costs [15]. It would take a long time to overcome these limitations one by one. Notably, almost all certified AMPs demonstrated equal or higher therapeutic efficacy than existing other commercial analogs or controls, as well as low toxicity to the host, a sensitive action to target pathogens, low/no residual tissue accumulation in tissues, and low resistance induction potential in bacteria [14]. It is known to all that the above merits as basic requirements are common among all new and old drugs.

The antibacterial activity of AMPs is influenced by a variety of structural parameters, such as structural tendency, hydrophobicity, amphiphilicity, and net charge. These parameters are interrelated, and antibacterial selectivity is the result of a delicate balance between them [41]. To explore the structure–function relationship of AMPs, the researchers mutated AMPs from natural sources or synthesized derivatives to achieve the goal of increasing antibacterial activity and reducing toxicity. Higgs et al. found that increasing the positive charge of chicken avian β-defensin-8 could significantly improve the antibacterial activity against *E. coli* and other four species of bacteria by targeting amino acid replacement. However, by increasing the charge of AMP with amino acid substitution at sites predicted to be positive for selection, the antimicrobial activity against *E. coli* was further enhanced. In contrast, there was no further increase in activity against the remaining pathogens [42]. The study results showed that net charge and the number of positively charged on the polar surface are important for both hemolysis and antimicrobial activity. The most striking observation was that the hemolysis changed dramatically with an increase in the positive charge of the V13K polar surface, i.e., the change from +8 to +9 led to a more than 32-fold increase in hemolytic activity [43]. In Chen’s study, the amphiphilic alpha-helical AMP V13KL served as a framework that systematically reduced or increased hydrophobicity through amino acid changes. The higher the hydrophobicity, the stronger the hemolytic activity. Reducing or increasing hydrophobicity beyond the optimal hydrophobic window significantly reduced antibacterial activity. Too strong hydrophobicity will lead to peptide self-polymerization and then affect its function [44]. There is a threshold between various physicochemical parameters of AMPs and antibacterial activity; it should be kept in a suitable range to avoid negative effects. In this study, we screened suitable mutants by introducing degenerate bases.

To overcome the high cost of new drug manufactured and low feasible competition with existing antibiotics, the mass production of AMPs at low cost is necessary and essential for their successful development, commercial deployment, and final usage in the population [45]. In 2003, Schnorr et al. [46] expressed Plectasin in *Aspergillus oryzae*, and the expression level was only about 50 mg/L. In 2010, Jing et al. [47] optimized the encoding sequence of Plectasin in an *E. coli* expression system and cloned it into the vector pET32a (+), which was expressed as a thioredoxin (Trx) fusion protein in *E. coli*. The fusion protein yield per liter of cell culture medium was about 92 mg. Only a 3.5 mg recombinant Plectasin polypeptide was obtained from 92 mg fusion protein. Above cases were far away from the feasible point let alone the desirable level. In 2011, Zhang et al. [37] in our team encoded Plectasin with a codon optimization gene, synthesized and cloned it into the pPICZaA expression vector, and expressed it successfully in the *Pichia pastoris* X-33 strain. After 120 h induction, the total amount of protein secreted by recombinant yeast reached 748.63 mg/L (supernatant). In the study, the yield of PN7 reached 5.53 g/L (supernatant) (Figure 5C), which was the highest expression level of similar AMPs at the laboratory level, which has significant importance for industrial scale in the future. It could be understood from the following points that one attributes the result to fermentation optimization during the many previous trials in our laboratory over the years. Another one is probably related to the mutation of three residues focusing on the N terminal of a helix structure as the sensitive region in a peptide. Some of the details (Figure 9) can be read out, i.e., D by Y with shifts, i.e., pI from 2.97 to 5.66 and Dalton from 133 to 181, and Q by R with a pI from 5.65 to 10.76 and Dalton from 146 to 174 with a longer base chain. It formed or made the mutant PN7 as new molecule with different physical and chemical characteristics from its parent peptide including pI from 7.77 to 8.61, an instability index from 13.82 to 19.50, and GRAVY from −0.695 to −0.722 (Table 2). The last one is probably related to the modification of post-transformation or post-transcription and shifting energy level for the biosynthesis of the target peptide during the recombination expression cycle in yeast cells, which was affected or changed from the above mutation and other relative events [48]. Its detailed mechanism should be further studied in the future.

Nuclear magnetic resonance spectroscopy (NMR) and X-ray crystallography assays showed that the spatial structure of Plectasin consists of three domains (N-terminal Loop, amphiphilic α-helix center, and two inverted parallel β-folds at the C-terminal), forming a typical CSαβ motif [32]. Compared to the Plectasin, the main structure of PN7 remains unchanged with a cysteine-stabilized α-helix/β-sheet motif (CSαβ) (Figure 9). As can be seen from Figure 10, the folding ratio of PN7 is higher, while the helical content of Plectasin is higher. That is consistent with the CD results (Table 4 and Table 5). The circular dichroism assay showed that the secondary structure of PN7 is mainly in β-folds (Table 4; Figure 5A). Both SDS and TFE provide a simple hydrophilic and hydrophobic interface similar to the membrane and can act as a membrane mimicry environment to induce an increase in the α-helical structure of PN7 (Table 4; Figure 5A), which may enable the peptide to keep stabilizing and interact without membranes of pathogens in the membrane medium.

AMPs have a wide range of highly effective bioactivities against fungi, protozoa, viruses, and tumor cells, so they have a wide application prospect and a solid attraction for end users. However, some scholars have pointed out that the broad-spectrum bioactivity of AMPs may lead to the imbalance of normal flora, causing serious side effects such as post-treatment complications [34,35,36]. In fact, our mutant PN7 has strong bactericidal activity only against G^+^, same as Plectasin (Table 6). In addition, the MIC value is an important factor in the initial selection of candidate AMPs and other drugs; a threshold of 16 μg/mL or 16 μmol/L MIC has been posed as a preliminarily entrance qualifying for clinical studies [17,49]. In this study, PN7 was shown to have antimicrobial activity against both *Staphylococcus* and *Streptococcus* at low MIC values of 1~16 μg/mL (Table 6). No obvious difference or conflict in antimicrobial activity and other characteristics was found between PN7 and plectasin. Let us look back at the last discussion description on the relationship between residue mutation and expression, the factors (Table 2; Figure 9) involving site residue mutation, helix, pI, GRAVY, unstable index, and CD shifts (Figure 5), and their possible effects on the antimicrobial mechanism and balance maintaining characteristics. The details that cannot be found yet in this work should be revealed continuously in the future.

The reason why AMPs failed in clinical trials is not only the failure to validate good activity consistent from in vitro models to in vivo models but also the toxicity [5]. In particular, some antibiotics, such as vancomycin and colistin, are known to have nephrotoxicity [50,51]. In order to confirm the peptide safety, we did in vivo and in vitro toxicity. PN7 showed almost no hemolysis (128 μg/mL: 0.19%) (Figure 6A) and a lower cytotoxicity (>85% survival) than plectasin (>60% survival) even at a high concentration of 128 μg/mL (Figure 6B,C). Meanwhile, after intraperitoneal injection of PN7 for 6 d, the mice showed no significant differences compared with the control group. These results imply that PN7 may be used to treat microbial infections without causing adverse effects (Figure 6D,E, Appendix A).

The instability of AMPs, usually contributing to weak druggability, is a major factor limiting their clinical usage. Many membrane-interfering AMPs employ α-helical amphiphilic structures to exert their antimicrobial activity. Polypeptide helices are characterized by multiple main hydrogen bonds between the carbonyl oxygen and amide NH that are continuously spirally rotating, and these interactions can vary depending on the local environment, such as ionic strength and pH [52]. It has been reported that the presence of cations can decrease or ablate the activity of some antimicrobial peptides, especially Ca^2+^ and Mg^2+^ [53,54,55]. After incubation of PN7 with different ions, it was found that the MIC value did not change probably because the ionic strength was weakened by dilution. However, when high ionic strength always exists in the reaction system, Ca^2+^ and Mg^2+^ increased really the MIC value by 2~8 times (Figure 7C,D). These results indicated that high concentrations of Ca^2+^ and Mg^2+^ could affect or inhibit the antibacterial activity of PN7. Further research is needed to determine whether cations affect the interaction between cationic peptides and anionic targets through simple charge competition or other ways. In this study, only the activity changes of PN7 were detected by dilution after incubation at different pH conditions for a period of time, and the activity in different pH environments should be evaluated scientifically in consistent reaction conditions or no dilution system, which would be more comprehensive but necessary. In addition, sensitivity to proteases leads to AMPs’ low bioavailability. The results showed that PN7 was sensitive to trypsin, resistant to pepsin, and stable in serum (Figure 7E,F).

In recent years, the time and concentration dependence of antibacterial drugs and post-effects have been increasingly valued in the treatment process to control drug resistance. Reasonable use is an important guarantee for achieving optimal efficacy and avoiding drug resistance, which provides feasible strong support for the rational selection of antibiotics in clinical practice in terms of both theoretical and practical dimensions [56]. The bactericidal curve can reflect the dynamic process and bactericidal efficiency of antibacterial drugs in killing bacteria. According to the characteristics of the bactericidal kinetic process, antibacterial drugs could be divided into two types: concentration-dependent or time-dependent antibacterial drugs (concentration-independent antibacterial drugs) [57]. As shown in Figure 8A, the slope of the curve for 1× MIC is negative, indicating that PN7 has a bactericidal effect and is a bactericide. Moreover, the negative value of the slope of the curve increases as the concentration increases, indicating that PN7 is a concentration-dependent antibacterial drug. Meanwhile, compared to vancomycin killing 99% MRSA ATCC 43300 in 8 h, a 2× MIC PN7 could completely kill bacteria and act rapidly just within 2 h, indicating that this rapid activity would contribute to the lower resistance and better development potential.

PAE, an important parameter for evaluating the pharmacodynamics of new antibacterial drugs, provides new ideas for the clinical design of more reasonable drug administration schemes and for drug production enterprises to design more reasonable drug formulations and dosage forms [58]. In this study, the PAEs of 1×, 2× and 4× MIC PN7 to *S. aureus* ATCC 43300 were 1.44 h, 1.84 h, and 2.82 h, respectively, which were superior to that of vancomycin (Figure 8B). The longer PAE of PN7 extends the interval time of administration in a clinical application; thus, reducing the frequency of administration will reduce the occurrence of adverse drug reactions, including residual and resistance, as well as the corresponding treatment costs.

It is crucial to understand how AMPs work in order to use them in clinical settings. Plectasin binds to lipid II to inhibit cell wall biosynthesis, thereby altering the bacterial morphology of *S. aureus* [33]. It is worth noting that the activity of AMPs is not limited to a single targeting mechanism; Many membrane-damaging cationic AMPs have secondary intracellular mechanisms of action [1,26,40]. P2 is derived from *Pyronema omphalodes* and has a similarity of 92% with plectasin, and it killed bacteria via interacting with plasma membrane by molecular oligomerization and destroying genomic DNA [40]. In the experiment, SEM assay showed that PN7 caused the surface of *S. aureus* ATCC 43300 to shape, crack and shrink, leading to bacterial rupture. Although it is difficult to determine the exact spatiotemporal relationship or complex causality among peptides, action precedence order with a target in detail, it is very interesting that the effect of PN7 on *S. aureus* ATCC 43300 was more rapid and more profound than that of Plectasin (Figure 8C), as observed by flow cytometry assay (Figure 8D). There is no use or the necessity to track its other intracellular event if the first target molecule of PN7, such as plectasin DO, is lipid II in the cell wall of the pathogen, even though many destroying events could be observed and validated in terms of intracellular molecular and cell levels. More importantly, a scientific spatiotemporal relationship or complex causality is the first basic stone or mark to disclose pharmacology during the characterization of action mechanisms for AMPs and other drug candidates, instead of the many indexes being rolled out or lined up regardless of their logistic relationship.

There is room to improve in some aspects in the future from this study. Firstly, how the amino acids at the 9th, 13th, and 14th sites in the sequence affect the properties of peptides should be revealed in subsequent experiments. Secondly, the action mechanism and target response of PN7 against target pathogens, i.e., spatiotemporal relationship or logistic causality, similar to or different from that of Plectasin, should be clarified or addressed firstly in the following experiments.

## 4. Materials and Methods

### 4.1. Construction of the Library

Multiple sequences (refer to previously reported research) [40,47,59,60,61,62,63] alignment and the sequence conservation analysis were performed using ClustalX 1.8 and the WebLogo (http://weblogo.berkeley.edu/logo.cgi) (accessed on 19 March 2023), and we introduced degenerate bases to different sites of amino acids.

The antimicrobial peptide template oligonucleotides encoding 40 amino acid sequences were synthesized with three, five, or seven NNN codons, where N represents A, T, G, and C. The PN gene sequences were confirmed by PCR amplification. The mixtures of ligated DNA were used to transform *E. coli* DH5α in order to amplify PN-DNA Library. The linearized recombinant plasmids pPIC-PNs were transformed into the *P. pastoris* X-33 cells for protein expression.

### 4.2. Screening of Active Clones

The experimental operation of the expression of the Plectasin variant in *P. pastoris* in well plates, shaking flasks, and 5 L level fermentation was performed as referenced in previous studies [26,40,64]. Candidates were identified through three different levels of screening. Firstly, a large number of transformants were screened through well plates to induce expression and determined by the inhibition zone assay (against *S. aureus* ATCC 43300) [26,65]. Secondly, the transformants with obvious inhibition zones were sequenced to obtain protein sequence information, then induced expression at the shake level. The supernatant was collected and purified to obtain the target protein; the anti-*Staphylococcus aureus* activity of the protein was detected by a MIC assay [24]. Finally, the codons of candidate sequences were optimized, the expression vector was reconstructed, the optimal transformers were obtained, and the high-density cultivation was performed in a 5 L fermenter. The final agent was determined by high protein yield and good activity (Figure 2).

### 4.3. Structure Analysis

The secondary structure of PN7 and Plectasin in different solvents was investigated by CD spectroscopy, which was performed as described in a previous study [40].

### 4.4. Antibacterial Activity (MIC)

The MIC value of PN7 against bacteria was determined by the microtiter broth dilution method [55]. Briefly, bacteria grown to the exponential phase were diluted into MHB; subsequently, 90 μL of bacterial suspension (10^5^ CFUs/mL) was mixed with 10 μL of different concentrations of PN7 in a sterilized 96-well microtiter plate (Nunc). MIC values were recorded after incubation for 16–18 h at 37 °C based on the lowest concentrations of peptides in which no visible bacterial growth was observed.

### 4.5. Safety Evaluation

We further evaluated the hemolysis, cytotoxicity to two mammalian cells, and toxicity to rodents about PN7.

The hemolytic activity of PN7 was evaluated based on the amount of the released hemoglobin from erythrocyte suspensions of healthy mouse blood [64]. Erythrocyte suspensions treated with 0.1% Triton X-100 and saline were used as 100% hemolysis and negative control, respectively.

The cytotoxicity of PN7 was evaluated by MTT (Beyotime, Shanghai, China) assay. PN7 was co-incubated with RAW264.7 cells or HaCaT cells for a period of time to detect whether it had an effect on cell growth [40].

ICR mice (6–8 weeks old, male, *n* = 5 per group; Charles River, Beijing, China) were intraperitoneally injected with PN7 (10 mg/kg, body weight) for 6 days, and the changes in mice behavior and body weight were recorded. At the same time, blood and organ tissues were collected for whole-blood cell and biochemical index analysis and H.E. staining [66].

### 4.6. Stability Analysis

#### 4.6.1. Thermal and pH Stability

PN7 dissolved in physiological saline solution was prepared into mother liquor with a concentration of 1280 μg/mL and incubated at different temperatures for 1 h. PN7 was dissolved in different pH buffers at 37 °C for 4 h. After double gradient dilution, MIC assays were used to detect the change in antibacterial activity [65].

#### 4.6.2. Salt Stability

The influence of ions on PN7 activity was determined in two ways.

The first method of operation: PN7 was prepared into 1280 μg/mL mother liquor supplemented with 50, 100, 200, 300, and 500 mM NaCl (or KCl, MgCl_2_, CaCl_2_), then was diluted twice gradient by ddH_2_O. Then, 10 μL was added into 90 μL bacterial solution and cultured for 16–18 h to determine the MIC.

The second method of operation: 10 μL peptide solution and 10 μL NaCl (or KCl, MgCl_2_, CaCl_2_) solution with different concentrations were added to 80 μL bacterial solution. The final concentration of ions in the whole system was 50, 100, 150, 200, and 250 mM. The MIC against *S. aureus* ATCC 43300 was determined.

#### 4.6.3. Serum and Proteolytic Stability

The stability of peptide PN7 in SGF, SIF, and 25% or 50% mouse serum was carried out as previously described [67]. A final concentration of 1280 μg/mL PN7 was prepared and incubated at 37 °C and then diluted twice. The MIC was determined.

### 4.7. In Vitro Bactericidal Kinetics

*S. aureus* ATCC 43300 was cultured to the logarithmic stage. MHB medium adjusted the concentration of bacterial suspension, and PN7 (1×, 2× and 4× MIC) was added into the suspensions containing 10^5^ CFU/mL bacteria, cultured at 37 °C and 250 rpm. 1×, 2×, 4× MIC Plectasin and 2× MIC vancomycin were used as positive controls, and the group treated without antibiotics was used as the blank controls. Then, 100 μL samples were taken at different time points and were continuously diluted and coated on MHA medium for colony counting [68,69]. A 100 CFU/mL is considered the detection limit.

### 4.8. PAE of PN7 against S. aureus

The post-antibiotic effect reflects the inhibitory effect on the growth of pathogenic bacteria after short contact with drugs, which is related to the strength of the bactericidal activity of drugs. The test method of the PAE of PN7 against *S. aureus* ATCC 43300 refers to the descriptions in previous studies [64,70].

### 4.9. SEM Observations

To further characterize the bactericidal effects of peptide, SEM was used to visualize the morphological changes. *S. aureus* ATCC 43300 cells were treated with PN7. Cells for SEM were processed as described previously [71].

### 4.10. Effect of PN7 on Membrane Penetrating

The fluorescent dye propidium iodide (PI; Solarbio, Beijing, China) was used to detect the effect of PN7 on the membrane permeability of *Staphylococcus aureus*. The test method refers to the descriptions in previous studies [72].

## 5. Conclusions

In summary, PN7 was screened in this study. The induced expression of PN7 was achieved at the level of a 5 L fermenter; after 120 h, the total protein concentration in the supernatant was 5.53 g/L, the highest record, as we know so far. PN7 had strong antimicrobial activity, especially regarding *Staphylococcus* and *Streptococcus*, and displayed low toxicity and high stability. Time–kill assays showed that PN7 acts rapidly and can inhibit the growth of *S. aureus* ATCC 43300 in a short time. The longer PAE would reduce administration times or extend the interval period. PN7 disrupted bacterial cell membranes. Our findings indicate that PN7 may be a novel promising antimicrobial agent. Moreover, this study also provides a model or an example for the design, modification, or reconstruction of novel AMPs and their derivatives.

## Figures and Tables

**Figure 1 antibiotics-12-01283-f001:**
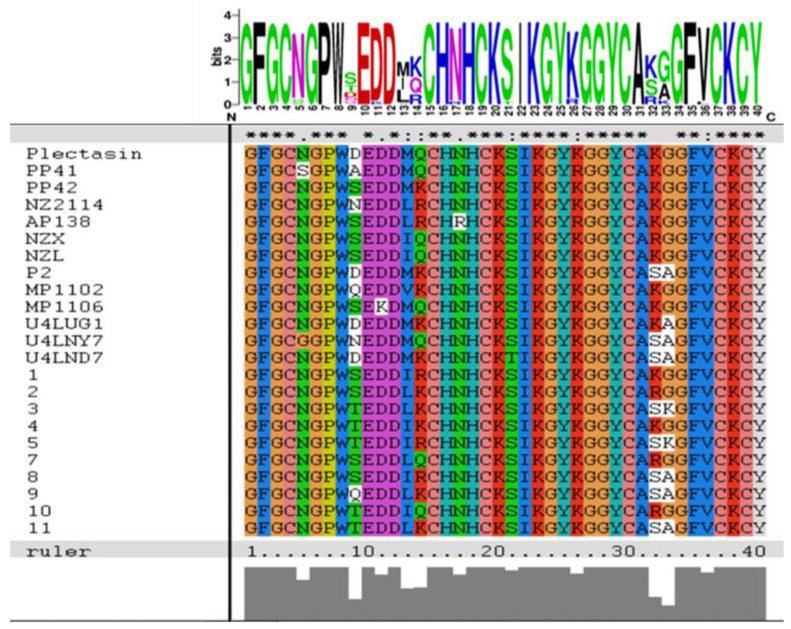
Sequence alignment using ClustalX 1.8 and WebLogo. The “*” indicates the conserved or unmutated amino acid residues. The blue and red represent basic amino acids and acid ones; the black color represent hydrophobic ones; the green and pink colors represent polar ones. The explanation for different colors is only for the upper part.

**Figure 2 antibiotics-12-01283-f002:**
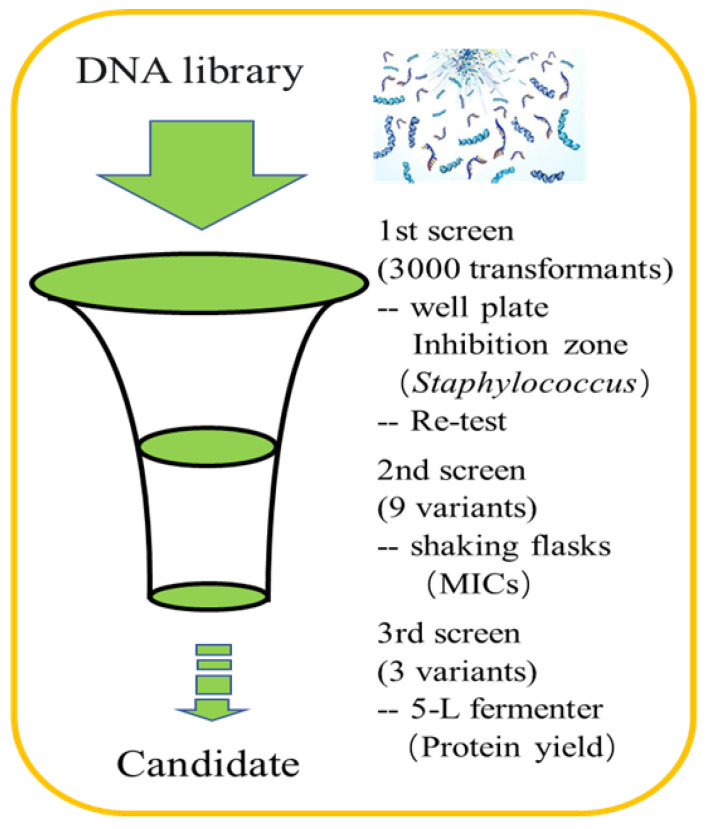
The screening process of the destination sequence. The study performed three rounds of inducible expression and screening at different levels: well plates, shake flasks, and 5 L fermenters. The detection methods of each round were the inhibitory zone, MIC, and protein yield, respectively. In total, 3000 active transformants were obtained by the inhibitory zone experiment. The transformants with good inhibitory zone effects were selected for sequencing, and nine sequences were determined. After the MIC assay, three candidates were performed for high-density expression in a 5 L fermenter.

**Figure 3 antibiotics-12-01283-f003:**

Part of the inhibition zones for well plate screening. The circle is marked as the positive control; the tick represents the screened good inhibitory zone effect transformants.

**Figure 4 antibiotics-12-01283-f004:**
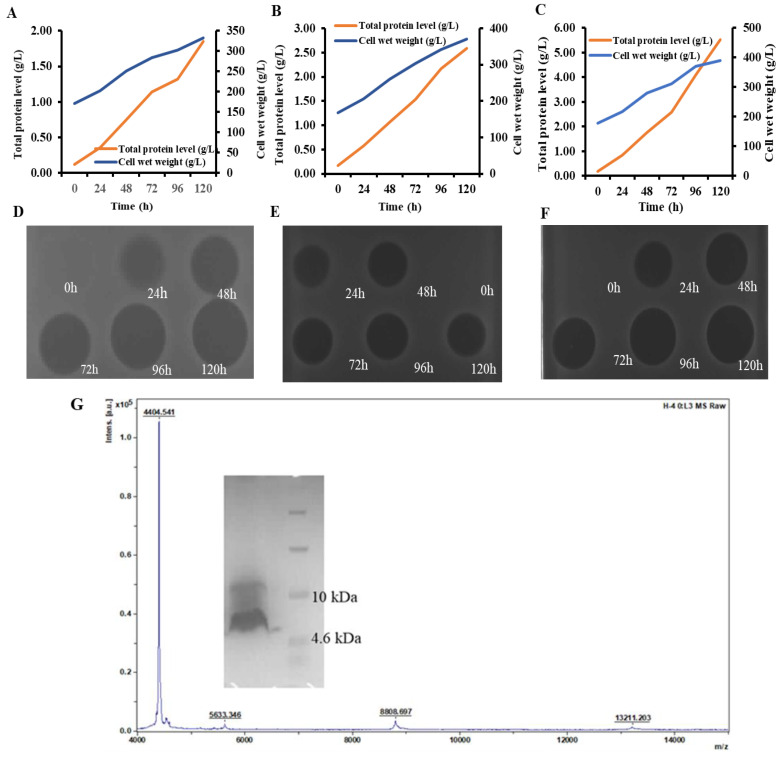
Expression and purification of the PNs in *P. pastoris* X-33. (**A**–**C**): Time curves of the cell wet weight and total protein yield during high-density fermentation of PN5, PN6, and PN7, respectively. (**D**–**F**): Bacteriostatic effect of the fermentation supernatant of PN5, PN6, and PN7, respectively. (**G**): Electrophoresis and mass spectrometry of PN7 after purification.

**Figure 5 antibiotics-12-01283-f005:**
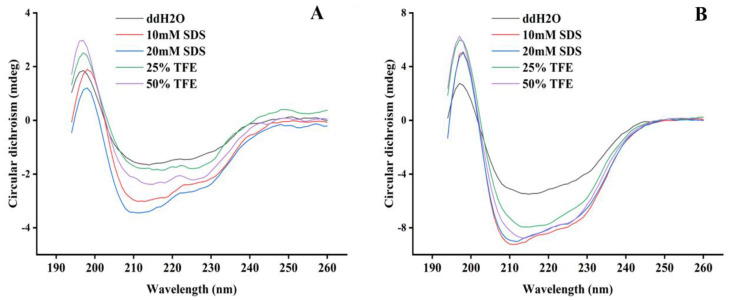
The CD spectra of the secondary structure of PN7 (**A**) and Plectasin (**B**).

**Figure 6 antibiotics-12-01283-f006:**
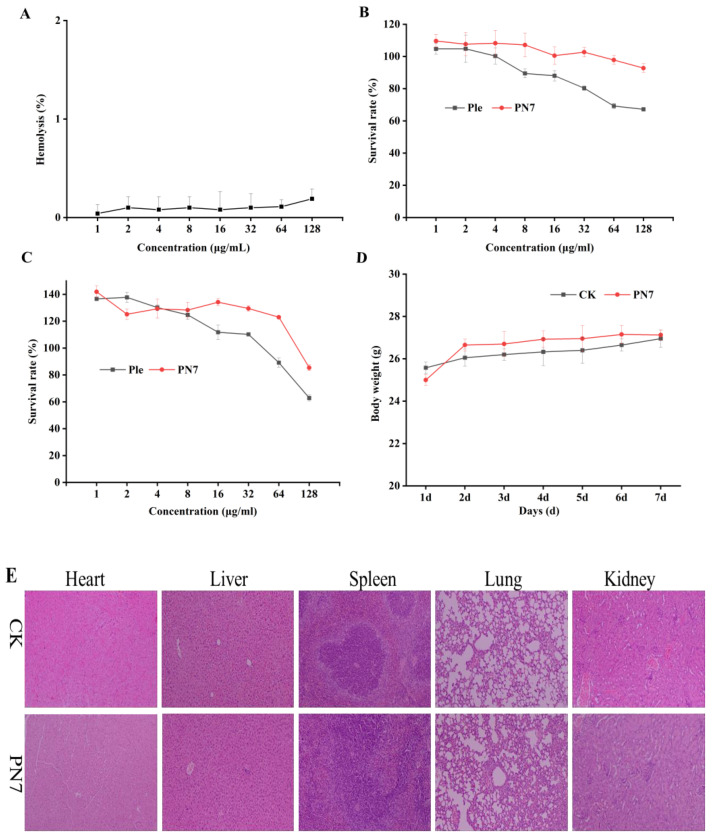
Safety of PN7. (**A**): Hemolytic activity of PN7 against fresh mice red blood cells; (**B**): cytotoxicity of the peptides against RAW264.7 cells; (**C**): cytotoxicity of the peptides against HaCaT cells; mice body weight (**D**) and histology images (H&E stained, (**E**)) of mice (ICR mice (6–8 weeks old, male, *n* = 5 per group; Charles River, Beijing) were intra-peritoneally injected with PN7 (10 mg/kg, body weight) for 6 days). Data are representative of three biological replicates, and the mean was shown. Ple stands for the Plectasin treatment group, and CK stands for the untreated group.

**Figure 7 antibiotics-12-01283-f007:**
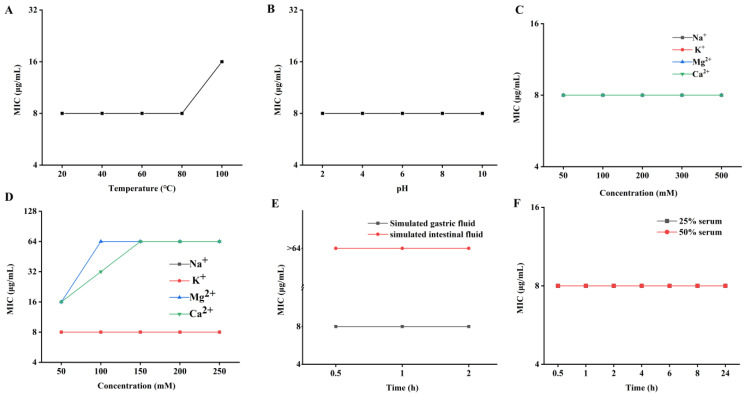
Stability of PN7 under different conditions. Effects of temperature (**A**), pH (**B**), salt ions (**C**, **D**), simulated gastric fluid and simulated intestinal fluid (**E**), and serum (**F**) on PN7 activity. All experiments were performed in triplicate.

**Figure 8 antibiotics-12-01283-f008:**
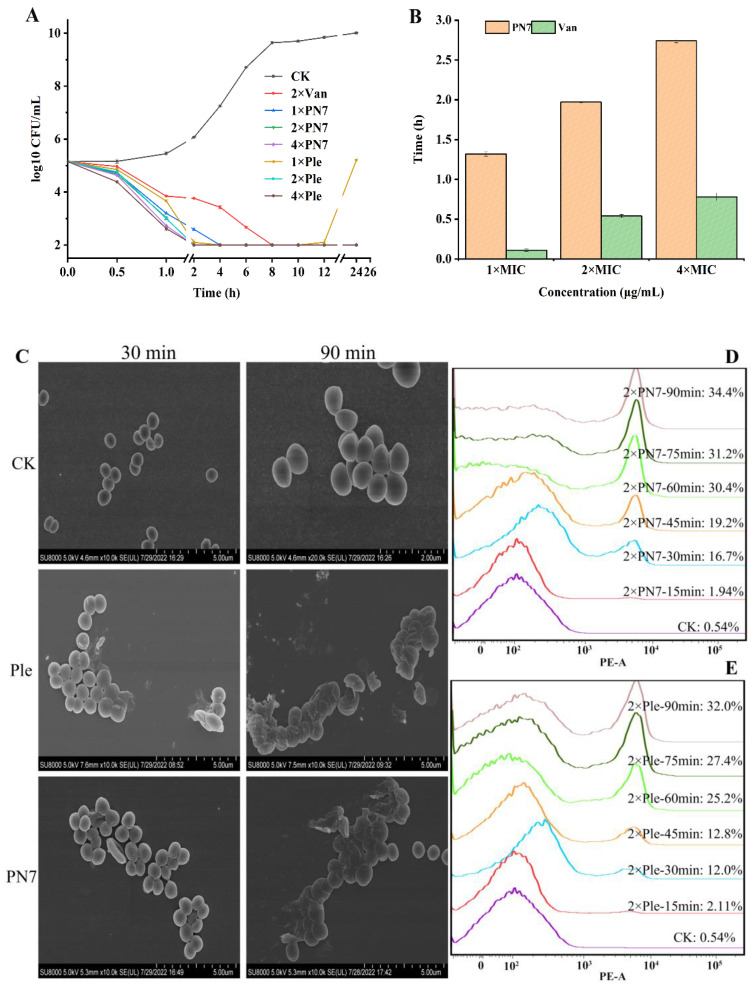
In vitro antimicrobial analysis of PN7. (**A**): Time-kill curve of PN7, Plectasin, and vancomycin against *S. aureus* ATCC. All experiments were performed in triplicate. (**B**): The PAE of 1×, 2× and 4× MIC of PN7 and vancomycin against *S. aureus* ATCC 43300 (the MIC values of PN7 and vancomycin against *S. aureus* ATCC 43300 were 8 μg/mL and 1μg/mL, respectively). All experiments were performed in triplicate. (**C**): The morphological influence of PN7 on *S. aureus* ATCC 43300 was observed by SEM. (**D**,**E**): The effects of PN7 (**D**) and Plectasin (**E**)) on membrane permeability were analyzed by FACS. The logarithmic phase *S. aureus* ATCC 43300 cells (10^8^ CFU/mL) were incubated with or without 2× MIC peptide solutions at 37 °C at different times. All experiments were performed in triplicate. Ple stands for the Plectasin treatment group, Van stands for the vancomycin treatment group, and CK stands for the untreated group.

**Figure 9 antibiotics-12-01283-f009:**
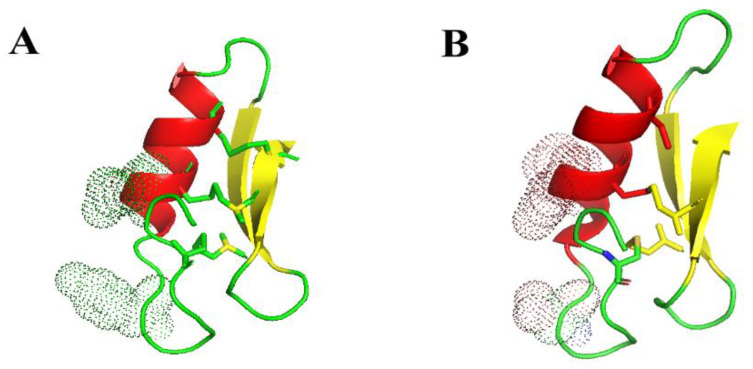
Conformation of the simulated higher structure of PN7 and its parental peptide. Note: The three-dimensional structures of PN7 and Plectasin were modeled using the SWISS-MODEL workspace (http://swissmodel.expasy.org/workspace/) (accessed on 17 June 2023) and PyMol 1.8. (**A**,**B**) refers to PN7 and Plectasin, respectively. Electron clouds around the mutation of three residues were marked in a peptide by discontinued green and brown point shapes, respectively. The red, yellow and green parts are mainly the different structures of the peptides, such as helix, sheet and random coli, respectively.

**Figure 10 antibiotics-12-01283-f010:**
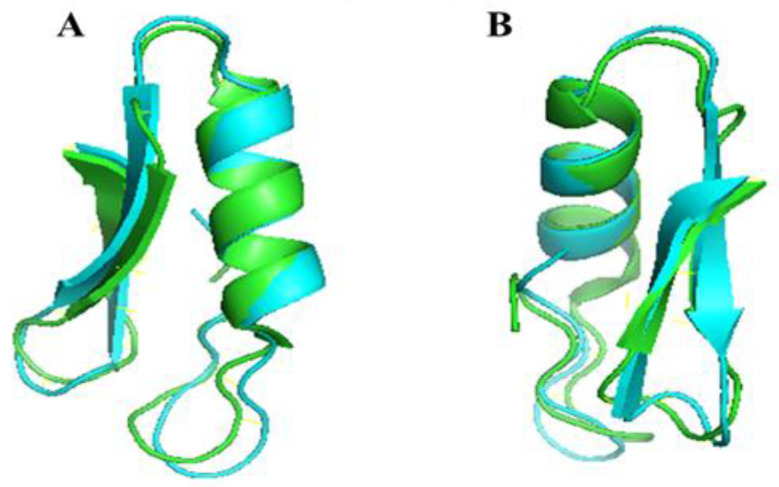
The superimposition of the structure of PN7 and Plectasin. The superimposition was produced using the PyMol software. (**A**,**B**) show different profiles. Blue stands for PN7, and green stands for Plectasin.

**Table 1 antibiotics-12-01283-t001:** Sequence design with degenerate bases.

ID	Sequence
PN-(3)	GGT TTT GGT TGT AAC GGT CCA TGG NNN GAA GAT GAT NNN NNN TGT CAT AAC CAT TGT AAG TCT ATT AAG GGT TAC AAG GGT GGT TAC TGT GCT AAG GGT GGT TTT GTT TGT AAG TGT TAC
PN-(5)	GGT TTT GGT TGT AAC GGT CCA TGG NNN GAA GAT GAT NNN NNN TGT CAT AAC CAT TGT AAG TCT ATT AAG GGT TAC AAG GGT GGT TAC TGT GCT NNN NNN GGT TTT GTT TGT AAG TGT TAC
PN-(7)	GGT TTT GGT TGT NNN GGT CCA TGG NNN GAA GAT GAT NNN NNN TGT CAT NNN CAT TGT AAG NNN ATT AAG GGT TAC AAG GGT GGT TAC TGT GCT AAG GGT GGT TTT NNN TGT AAG TGT TAC

**Table 2 antibiotics-12-01283-t002:** Sequences and physicochemical properties of Plectasin and PN1–PN9.

Name	M.W.	PI	Charge	GRAVY	Instability Index	Sequences
Plec	4407.99	7.77	+1	−0.695	13.82	GFGCNGPWDEDDMQCHNHCKSIKGYKGGYCAKGGFVCKCY
PN1	4274.87	8.30	+2	−0.438	20.68	GFGCNGPWLEDDAGCHNHCKSIKGYKGGYCAKGGFVCKCY
PN2	4347.92	8.62	+3	−0.708	11.71	GFGCNGPWREDDTGCHNHCKSIKGYKGGYCAKGGFVCKCY
PN3	4447.06	8.86	+4	−0.810	15.71	GFGCNGPWREDDRTCHNHCKSIKGYKGGYCAKGGFVCKCY
PN4	4318.92	8.30	+2	−0.428	28.58	GFGCNGPWIEDDATCHNHCKSIKGYKGGYCAKGGFVCKCY
PN5	4347.71	8.30	+2	−0.568	26.75	GFGCNGPWNEDDVTCHNHCKSIKGYKGGYCAKGGFVCKCY
PN6	4479.11	8.30	+2	−0.630	19.50	GFGCNGPWWEDDMQCHNHCKSIKGYKGGYCAKGGFVCKCY
PN7	4409.99	8.61	+3	−0.722	19.50	GFGCNGPWYEDDGRCHNHCKSIKGYKGGYCAKGGFVCKCY
PN8	4332.91	8.61	+3	−0.762	18.56	GFGCNGPWNEDDGKCHNHCKSIKGYKGGYCAKGGFVCKCY
PN9	4333.89	8.62	+3	−0.710	19.50	GFGCNGPWREDDGSCHNHCKSIKGYKGGYCAKGGFVCKCY

**Table 3 antibiotics-12-01283-t003:** The MIC of the PN1–PN9 protein induced by shaking flasks against *Staphylococcus aureus*.

Peptide	MIC (μg/mL)			
	*S. aureus* 43300	*S. aureus* 25923	*S. aureus* E48	*S. aureus* 546
PN1	8	16	4	8
PN2	8	16	4	8
PN3	8	16	8	8
PN4	16	32	4	8
PN5	4	16	4	4
PN6	8	16	8	8
PN7	4	16	4	4
PN8	16	>32	4	16
PN9	16	>32	>16	>16

**Table 4 antibiotics-12-01283-t004:** Analysis of the secondary structure of PN7 in different solutions.

Secondary Structure	The Percentage of the Secondary Structure in Different Solvents (%)				
	PN7-H_2_O	PN7-10 mM SDS	PN7-20 mM SDS	PN7-25% TFE	PN7-50% TFE
Helix	7.63	8.43	8.71	8.04	8.76
Antiparallel	41.39	35.91	33.03	40.10	39.17
Parallel	3.72	3.81	3.65	3.82	3.94
β-turn	17.22	18.25	19.45	17.35	17.52
Random coli	30.04	33.60	35.16	30.69	30.51

**Table 5 antibiotics-12-01283-t005:** Analysis of the secondary structure of Plectasin in different solutions.

Secondary Structure	The Percentage of the Secondary Structure in Different Solvents (%)				
	Ple-H_2_O	Ple-10 mM SDS	Ple-20 mM SDS	Ple-25% TFE	Ple-50% TFE
Helix	15.4	17.6	17.2	19.3	19.9
Antiparallel	29.9	25.8	27.1	24.5	22.3
Parallel	4.7	5.1	5.0	5.5	5.6
β-turn	18.9	19.7	19.9	17.1	18.9
Random coli	31.0	32.0	31.5	32.9	32.8

**Table 6 antibiotics-12-01283-t006:** Antibacterial spectrum (MIC).

Strains	MIC (μg/mL)		Source
	PN7	Plectasin	
Gram-positive bacteria			
*Staphylococcus aureus* ATCC 43300	8	4 ^a^	ATCC
*S. aureus* ATCC 25923	16	NT	ATCC
*S. aureus* CVCC 546	1	16 ^a^	CVCC
*S. aureus* E48	1	4 ^a^	Northwest A&F University
*S. aureus* CICC 10473	4	NT	CICC
*S. epidermidis* ATCC 35984	16	16	ATCC
*S. hyicus* 437-2	8	NT	Tianjin Institute of Animal Sciences
*S. hyicus* NCTC 10350	4	NT	NCTC
*Streptococcus. suis* CVCC 606	1	1 ^a^	CVCC
*S. agalactiae* ATCC 13813	2	2	ATCC
*S. agalactiae* CAU-FRI-4	1	NT	Clinical strain (the laboratory)
Gram-negative bacteria			
*Escherichia coli* CVCC 195	>128	>128 ^a^	CVCC
*E. coli* CVCC 1515	>128	NT	CVCC
*E. coli* O157	>128	>128 ^a^	CVCC
*E. coli* ATCC 25922	>128	NT	ATCC
*Salmonella enterica* ATCC 13076	>128	NT	ATCC
*S. enteritidis* CVCC 3377	>128	NT	CVCC
*S. pullorum* CVCC 1789	>128	>128 ^a^	CVCC
*Pseudomonas aeruginosa* CICC 20625	>128	NT	CICC
*P. aeruginosa* CICC 21630	>128	>128 ^a^	CICC
Fungus			
*Candida albicans* CICC 98001	>128	>128	CICC

NT: no test. ^a^: Data cited from previous results [40].

## Data Availability

The original contributions presented in the study are included in the article; further inquiries can be directed to the corresponding author(s).

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
