# Peer review of "Site Mutation Improves the Expression and Antimicrobial Properties of Fungal Defense"

_antibiotics, 2023, doi:10.3390/antibiotics12081283_

Round 1

Reviewer 1 Report

1.      The authors should double-check the title for correctness. Is it a fungal defensin or a fugal defense?

2.      The title appears to be vague; I recommend that authors create a title that details the investigations in particular.

3.      I recommend that the authors produce a superimposition of PN7 and its parental peptide and discuss any significant structural differences.

4.      The authors should discuss the literature/study reports regarding the relationship between mutations and toxicity, stability, and activity profile of AMPs.

5.      Have the authors conducted any target experiments on this new mutant agent?

6.      What about this new agent's antifungal activity?

Reviewer 2 Report

The authors aim at improving the activity and production of plectasin by mutagenesis and found a promising candidate peptide called PN7. Such discovery is of great interest and timely in the fight against antibiotic resistant S. aureus. However, this manuscript text is barely readable with many incomplete or meaningless sentences, many typos, etc. Figures cannot be understood as they lack proper legends, contain incomplete axes titles, etc. Results are described without a proper introduction or the necessary information to understand the methods used and to assess their relevance. Methods are poorly described or absent, with for only information a reference from the literature. Results from figures 8A and 8D-E are contradictory. These multiple problems make it hard to judge on the scientific quality and relevance of the paper. I would advice the authors to revise the manuscript entirely and resubmit it in a more polished form. Nevertheless, here are several comments on the manuscript.

Title: fungal instead of fugal

Lines 27-28. The sentence is not clear, please rephrase. Do the authors mean: PN7 has good potency against antibiotic-resistant gram+ bacteria?

Lines 30-31 The sentence is not clear, please rephrase. What do the authors mean with “PN7 kills S. aureus with mode of more quickly”?

Lines 45-46. Could the authors elaborate on the fact that AMPs killing mechanisms makes bacteria less susceptible to drug resistance? From literature, it appears that on the contrary, resistances occur very quickly against most bacteriocins as an example, but not from some compounds like teixobactin.

Lines 49-52. The sentence is not clear, please rephrase. Could the authors also provide a few examples of AMPs that have successfully gone through clinical trials?

Lines 70-71. Why is the druggability of animal-derived AMPs weak and not the microbial-derived ones? Please elaborate on this.

Lines 72-74. Why and how are microbial-derived peptides similar to antibiotics?

Lines 75-78. Please describe in sufficient details the mechanism of action of plectasin as this is central to the paper.

Lines 94-95. Remove “such as”

Lines 94. What sequence alignment are the authors referring to? Please introduce the experiment that was done and why it was done for more clarity.

Lines 99-105. It is not clear how the screening was performed and on what basis the 9 candidates were isolated. Please provide more information on the rationale behind the screening and the different sequential steps. In addition, Figure 2 shows only 3 candidates after the 3 screening steps, why 9 in the text?

Figure 2. What is the meaning of “shaking flasks vs MICs”? Please explain this more in detail to improve the clarity of the figure.

Figure 3. The figure lacks a legend and a proper description.

Lines 120-128. Please explain CD spectra and do not use abbreviations without defining them (circular dichroism spectroscopy should appear at least once).

Lines 131. Please provide the real MIC values instead of a range.

Lines 134. Does PN7 affect all gram+ as it is suggested above in the text? If this is the case, I wouldn’t call it precision therapy, please rephrase the sentence accordingly.

Figure 6. The figure lacks a legend and a proper description. Panels are not described at all. Methods are lacking as well. Font size is too small and barely readable on tables and figures.

Lines 140-141. Lower than should be replaced by higher than

Line 150. What table?

Lines 162-173. How is the treatment done, are bacteria in stationary phase or exponential phase, please provide the necessary information to understand the results depicted here.

Figure 8A. Please explain why addition of antimicrobials/killing is done at 10^5 CFU/mL. This is an extremely low concentration of bacteria, equivalent to approximately OD600 0.001. This also reduces the resolution of the assay as authors stop measuring CFUs/mL at 10^2 CFU/mL. Please repeat this assay with a more standard protocol or elaborate on the choice of this protocol. Please also measure CFU/mL down to 10 CFU/mL to increase the resolution of the graph.

Figure 8B. What does concentration refer to?

Figure 8D-E. PI positive cells go up to a maximum of 32-34% of the population. This implies that only 32-34% of cells undergo membrane permeabilization. However, the killing curves indicate a killing of at least 99.9% of the cells. How can the authors reconciliate these results?

Figure 8C. SEM images are not described in the results section.

Included above

Reviewer 3 Report

Hao et al. screened the mutation library of Plectasin and found the PN7 variant. The PN7 variant gives a higher yield in the culture supernatant. PN7 shows antimicrobial activity against Staphylococcus aureus, stability in different environmental conditions, and low cytotoxicity. This is very convincing work, but it lacks a comparison study with Plectasin. Putting data from a comparison study into this study will enhance it, especially for antimicrobial assays and structural studies. Following are specific suggestions for this study:

Figure 3: Please describe the figure legend and detail for the plates.  

Figure 4, G: Values in mass spectrometry data are not visible. Lines 117–119, explain why there is an addition of ~5 Da. Is there any additional modification in purified peptide? Author should provide MS-MS for the same peptide so that amino-acid sequence changes can be seen. Authors should also include wild-type peptide for MS-MS analysis.

Lines 120–128, Structure Analysis: I think authors should include wild-type peptide so that readers will understand the structural similarity or differences with respect to wild-type.

Lines 129–133, Table 5: Authors should provide representative data in figure format with statistical significance and include wild-type peptide as a positive control.

Figure 6: Please describe CK and ple used in the figure.

Lines 149–159, Desirable stability of PN7: Authors should mention how many times experiments were repeated and also put SD in figures.

Figure 8: Section A; please include SD and mention the repetition of the experiment in the figure legend. Section D and E graph values are not visible.

Figure 9 is not described in the result section.

Round 2

Reviewer 1 Report

Please verify and maintain the uniformity of the titles of the manuscript and supplementary files. Moreover, the authors have completed the majority of the modifications I recommended. Now, the manuscript is eligible for publication consideration.